# Dendritic Cell-Based Vaccines Recruit Neutrophils to the Local Draining Lymph Nodes to Prime Natural Killer Cell Responses

**DOI:** 10.3390/cells12010121

**Published:** 2022-12-28

**Authors:** Lily Chan, Yeganeh Mehrani, Geoffrey A. Wood, Byram W. Bridle, Khalil Karimi

**Affiliations:** 1Department of Pathobiology, Ontario Veterinary College, University of Guelph, Guelph, ON N1G 2W1, Canada; 2Department of Clinical Science, School of Veterinary Medicine, Ferdowsi University of Mashhad, Mashhad 91779-48974, Iran; 3ImmunoCeutica Inc., Cambridge, ON N1T 1N6, Canada

**Keywords:** neutrophils, dendritic cell vaccine, crosstalk, immunobiology, natural killer cells

## Abstract

Dendritic cell (DC)-based cancer vaccines are a form of immunotherapy that activates the innate and adaptive immune systems to combat cancers. Neutrophils contribute to cancer biology and have the potential to be exploited by immunotherapeutic platforms to enhance anti-tumor immune responses. We previously showed that DC vaccines elicit the expansion of mouse interferon (IFN)γ-producing mature natural killer (NK) cells to elevate anti-tumor responses. Here, we demonstrate the rapid recruitment of neutrophils to the draining lymph nodes of DC-vaccinated mice. This was accompanied by an increase in the total number of NK cells producing IFNγ and expressing CD107a, a marker of degranulation that demonstrates NK cell functional activity. Furthermore, the depletion of neutrophils in DC-immunized mice resulted in decreased numbers of NK cells in draining lymph nodes compared to the controls. Interestingly, the increased number of IFNγ- and CD107a-expressing NK cells in DC-immunized mice was not detected in mice depleted of neutrophils. Further investigations showed that DC vaccines induced IFNγ− and TNFα-producing CD8+ T cells that also expressed CD107a, but depletion of neutrophils did not have any impact on the CD8+ T cell population. Our findings suggest that neutrophil-mediated anti-tumor immunity induced by a DC vaccine platform could be targeted to provide innovative strategies to enhance its clinical efficacy.

## 1. Introduction

Dendritic cell (DC) vaccines are an immunotherapy that typically involves the use of isolated DCs from a patient or the *ex vivo* generation of DCs from patient-derived progenitor cells to activate the patient’s immune system and stimulate antigen-specific responses [1]. Immunotherapy relies on the functions of DCs. DCs are components of the innate immune system that are critical players in both innate and adaptive immunity and are thought of as the interface between the two. They are professional antigen-presenting cells, making them critical in the education of the adaptive arm of the immune system [2]. DC vaccines allow for the induction of desired antigen-specific responses through the *ex vivo* manipulation of DCs. DC vaccines can be loaded with tumor-specific antigens and, once administered to the patient, will stimulate and promote tumor-directed immune responses [3]. Immunotherapies such as DC vaccines are amongst the new era of personalized medicine. DC vaccines allow for the use of autologous cells for vaccine preparation and patient-specific antigens [3]. Although DC vaccines hold a significant amount of promise, there has been limited efficacy observed in clinical trials [4]. Therefore, there is a need to enhance the understanding of the immunobiology of DC vaccines to improve the development and generation of DC-based immunotherapies. This includes investigating the communications occurring in the immune system following DC vaccination. The roles of T cells, specifically cytotoxic T cells, in DC vaccination has been a key area of research since the premise of DC vaccines was founded on the critical role of DCs in educating these cytotoxic effector T cells [2]. Natural killer (NK) cells are cytotoxic cells of the innate immune system that are essential effector cells in anti-cancer immunity. We and others have also demonstrated important roles of DC vaccines in inducing anti-tumor responses that involve NK cells [5,6]. However, there has been limited research into the crosstalk occurring between DC vaccines and other components of the innate immune system. This includes the first responder leukocytes known as neutrophils. 

Neutrophils are phagocytic leukocytes that are abundant in blood. They are known as immediate responders because they rapidly accumulate at sites of injury and infection. They are granulocytes that mediate killing by producing reactive oxygen species, antimicrobial peptides, and neutrophil extracellular traps [7]. Although conventionally thought of as rapid responders that facilitate the killing and clearance of pathogens and debris, neutrophils have also been shown to have more extensive roles, including activation of other innate responses and involvement in the education of adaptive immunity [8,9,10,11,12]. Neutrophils can contribute to the induction of adaptive immunity both directly and indirectly. For instance, they have been shown to have antigen-presenting capacities [11,13]. Furthermore, as primary responders and typically one of the first leukocytes to traffic to sites of inflammation or injury, they have essential roles in the recruitment of other leukocytes [14]. 

Neutrophils display heterogenicity and plasticity, which can be influenced by environmental stimuli [15,16,17]. As much as the immune system and its components are designed to be protective, there is also a capacity for harm. Neutrophils have been implicated in harmful immunological processes [18,19,20]. Like macrophages, neutrophils can display polarized phenotypes in the tumor microenvironment. These N1 and N2 cells demonstrate plasticity that can be influenced by external stimuli [21]. Thus, neutrophils can walk a fine line between being key players in immunological defenses while also having detrimental contributions to pathological conditions. This dual nature in contributions to both health and disease insinuates a potential to manipulate neutrophil phenotypes via immunotherapeutic tools to obtain desired outcomes. There has been limited research investigating the roles of neutrophils in DC-based vaccine immunobiology. However, DCs and neutrophils have established communications and crosstalk in various immunological contexts beyond DC-based vaccines [22,23,24,25]. The hypothesis tested here was that neutrophils are involved in the immune responses elicited by DC-based cancer vaccines. Using murine models, it was discovered that a population of neutrophils rapidly accumulated in vaccine-draining lymph nodes. Using antibody-mediated depletion, the potential roles of neutrophils in the migration of the DC-based vaccine and the concomitant activation of NK cell responses were examined. 

## 2. Materials and Methods

### 2.1. Mice

Specific pathogen-free female C57/Bl6 mice aged 36–52 days were obtained from Charles River Laboratories, Quebec, Canada. Upon arrival at the Animal Isolation Unit at the University of Guelph, Guelph, Ontario, Canada, the mice were given one week to acclimatize to their new environment before experiments were started. Mice were housed in a controlled environment with food and water provided *ad libitum*. All mouse-based studies were approved by the University of Guelph’s Animal Care Committee under Animal Utilization Protocol #3807 and complied with the guidelines of the Canadian Council on Animal Care.

### 2.2. Dendritic Cell Culture 

Tibias and femurs were harvested from euthanized mice. The ends of the femurs and tibias were cut off, and the bone marrow was flushed into a Petri dish with phosphate-buffered saline (PBS) using a 23G × 3/4 needle. The bone marrow was resuspended into a single cell suspension and passed through a filter with a 70 µm pore size into a 50 mL conical tube. The cells were counted and resuspended to 1.25 × 10^6^ cells/mL in culture media (1% penicillin/streptomycin, 10% fetal bovine serum (VWR, Mississauga, ON, Canada, Cat#97068-085), and 0.1% 2-beta-mercaptoethanol in RPMI). Granulocyte-macrophage colony-stimulating factor (GM-CSF) (Biolegend, BioLegend, San Diego, CA, USA, Cat#576308) was added to a concentration of 20 ng/mL. The cells were then plated in 25 cm^2^ flasks on day zero of the cultures and placed in a 37 °C humidified incubator with 5% CO_2_. Media and GM-CSF were supplemented on day two and day five of the cultures. On day two 5 mL of media with 20 ng/mL GMCSF was added. On day five half 5 mL was removed from each culture, centrifuged, and the supernatant was removed. The cells were resuspended in fresh media with 20ng/mL GMCSF and added back to the cultures. On day seven, the cultures were harvested and used to perform experiments.

### 2.3. DC Vaccine Preparation

On day seven of the DC cultures, the cultures were transferred from their 25 cm^2^ flasks to 50 mL conical tubes, and the cells were counted using a hemocytometer. The cells were stimulated for one hour with lipopolysaccharide (LPS) from Escherichia coli O55:B5 (Sigma, Oakville, Canada, Cat#L2880) at a concentration of 100 ng/mL and 1 µg/mL of both chicken ovalbumin (OVA)_257-264_ (SIINFEKL or SIIN) and chicken and OVA_323-339_ (ISQAVHAAHAEINEA or ISQ) peptides (both from PepScan Systems, Lelystad, The Netherlands), which are immunodominant epitopes for CD8^+^ and CD4^+^ T cells, respectively, in mice with major histocompatibility complex haplotype ‘b’. During the stimulation, the cells were placed in the incubator at 37 °C and the tubes were mixed by swirling every fifteen minutes. Following stimulation, the cells were washed with three times with PBS and then resuspended in PBS to a concentration of 1 million cells per 60 µL. Each vaccine dose was 0.5 million cells in 30 µL of PBS, administered into the footpad of a hind limb (Appendix A).

### 2.4. Processing Spleens and Lymph Nodes

Spleens and lymph nodes were harvested from euthanized mice. Spleens and lymph nodes were placed into Petri dishes containing Hanks buffered saline solution (HBSS) and pressed into single cell suspensions using the plunger from a syringe. Suspended cells were passed through a filter with a 70 µm pore size into a 50 mL conical tube. Erythrocytes were lysed using ACK lysing buffer (1 g KHCO_3_ [10.0 mM], 8.29 g NH_4_Cl [0.15 M], 37.2 mg Na_2_EDTA [0.1 mM] in 1 mL of Milli Q water). Cells were resuspended in HBSS. 

### 2.5. NK and T Cell Response Assays

Splenocytes or lymph node-derived cells were resuspended in media (RMPI with 10% FBS, 1% penicillin/streptomycin, 0.1% 2-beta-mercaptoethanol) containing a CD107a-specific antibody (eBioscience, San Diego, CA, USA, Cat#46-1071-82) to detect degranulation, and plated into two wells, one which served as an unstimulated control well and the other as a stimulated test well. For assessment of NK cells, they were non-specifically stimulated with 100 ng/mL of interleukin-2 (Biolegend Cat#575406). For assessment of T cells, they were re-stimulated with the same two peptides that had been loaded onto the DC vaccine, each at a concentration of 1 µg/mL. Cells were incubated for one hour at 37 °C before brefeldin A (Biolegend Cat#420601) was added to a final concentration of 1000× *g* and incubated for another four hours at 37 °C. The cells were washed with PBS and stained for intracellular cytokine analysis.

### 2.6. Antibody Staining 

Splenocytes were resuspended in PBS and 1 million cells per well were seeded into 96-well plates. The lymph nodes were seeded into 96-well plates based on the total counts obtained after processing. The samples were plated and then cells were resuspended in anti-CD16/CD32 (BioLegend Cat#101320) to block Fc receptors and incubated for 15 min at 4 °C. The cells were washed twice with PBS and resuspended in a mastermix of surface-staining antibodies (Ly6G [BD Pharmingen Cat#551461], CD11b [Biolegend Cat#101206], NK1.1 [BD Pharmingen, Oakville, Canada, Cat#550627], CD3ε [Biolegend Cat#100336], CD4 [eBioscience Cat#25-0042-82], CD8a [eBioscience Cat#11-0081-86]) and incubated at 4 °C for 20 min. The cells were washed twice with FACS buffer (0.5% bovine serum albumin [HyClone, Ottawa, Canada, Cat# SH30574.02] in PBS) twice and then resuspended in a fixable viability dye (FVD) (BioLegend Cat#423101) and incubated for 25 min. The cells were washed twice with FACS buffer and analyzed via flow cytometry. The samples were run whole unless the counts were over 1 million cells. If the samples had over 1 million cells, they were seeded to be 1 million. Neutrophils (Ly6G^+^ CD11b^+^), NK cells (CD3^−^ NK1.1^+^), CD8+ T cells (CD3^+^ CD8^+^), and CD4+ T cells (CD3^+^ CD4^+^) were assessed using surface antibody staining (Appendix A). 

### 2.7. Intracellular Cytokine Staining

In some experiments, assessments of cytokine expression were conducted. Following surface antibody and FVD staining, cells were fixed by incubating with a permeabilization and fixation buffer (BioLegend Cat#420801) for 10 min. Cells were washed twice with permwash buffer (BioLegend Cat#421002) and then resuspended in a mastermix of cytokine-detecting antibodies containing anti-TNF-α (BioLegend Cat#506308) and anti-IFNγ (BioLegend Cat#505808) and incubated for 20 min at 4 °C. Cells were washed twice in permwash buffer and resuspended in FACS buffer to be run through the BD FACSCanto^TM^ II flow cytometer and analyzed using FlowJo software v10 (Ashland, OR, USA). 

### 2.8. Ly6G Depletion

Mice in the Ly6G depletion treatment group were given doses of 250 ug of *InVivoPlus* anti- mouse Ly6G (Bio X Cell, Burlington, Canada) in 200 uL PBS via intraperitoneal injection the day before DC immunization, the day of immunization, and every 2–3 days throughout the induction phase of the immune response.

### 2.9. Staining with Carboxyfluorescein Succinimidyl Ester (CFSE)

For *ex vivo* analysis of DCs trafficking from the injection site to the draining lymph node, cells were stained with the tracking dye CFSE. Cells were washed with PBS and resuspended at a concentration of 10 million cells per mL of PBS in a 15 mL conical tube. CFSE was added at a concentration of 1 µM and vortexed immediately and incubated in a water bath at 37 °C for 15 min. The cells were then diluted ten-fold with PBS and centrifuged for five minutes at 500× *g*. The supernatant was removed, and cells were washed with 10 mL of media (1% penicillin/streptomycin, 10% fetal bovine serum, 0.1% betamercaptoethanol in RPMI). Cells were then washed with PBS twice and resuspended for immunizations. 

### 2.10. Statistical Analyses 

All graphing and statistical analyses were performed with GraphPad Prism 9 software (GraphPad Software, Inc., San Diego, CA, USA). Graphs display means and standard errors. Data were analyzed using a two-way analysis of variance (ANOVA), one-way ANOVA, or unpaired *t*-tests when there were two variables, multiple iterations of a single variable, or two treatment groups, respectively. Treatments were considered significantly different from controls if the *p*-value was ≤0.05.

## 3. Results 

First, the ability of the DC vaccine to influence the cellularity of the lymph node draining the immunization site was investigated. Following DC vaccination into the footpads of mice, changes in the total number and proportion of T cells and NK cells populations were observed in the local draining lymph node (Figure 1). This established that the DC vaccine induced cellular changes in the lymphatic system. Since NK and T cells have been a central focus of DC vaccine immunobiology, we expanded our investigation to determine if there were changes in understudied leukocyte populations.

A population of Ly6G^+^ cells appeared in the lymph node shortly following DC vaccination. A kinetics experiment was performed to examine the population. The Ly6G^+^ population appeared to peak around 12 h post-DC immunization and quickly declined in numbers and was undetectable five days post-vaccination (Figure 2). 

The recruitment of the Ly6G^+^ population was further investigated to determine if it was limited to the draining lymph node. Therefore, the vaccine-draining popliteal lymph node, the non-vaccine-draining popliteal lymph node, and the inguinal lymph node were examined for the presence of a Ly6G^+^ population. The non-draining inguinal lymph node had similar results as the lymph node from PBS-treated control mice, indicating that the accumulation of neutrophils following DC vaccination is limited to the local vaccine-draining lymph node (Figure 3). 

Ly6G is a common marker for identifying neutrophils in mice. Therefore, the vaccine-draining lymph nodes were examined six-, 24-, and 48 h post-DC immunization to see if the Ly6G+ population could be attributed to neutrophils (Ly6G+ CD11b+). A population of neutrophils was observed in the vaccine-draining lymph nodes that were not present in the PBS-draining popliteal lymph nodes or in non-draining inguinal lymph nodes from the vaccinated side of the mice (Figure 4). 

To investigate the nature of the DC vaccine-induced accumulation of neutrophils in the draining lymph nodes, DCs were prepared in multiple ways. Specifically, different steps of the vaccine preparation were excluded to help elucidate the mechanisms behind the signaling. PBS was used as a negative control. Treatment groups received the full DC vaccine that had been stimulated with lipopolysaccharide (LPS) and were antigen-loaded (DC + LPS + Ag), DCs that were not stimulated but antigen-loaded (DC + Ag), DCs that were stimulated with LPS (DC + LPS) but that were not antigen-loaded, and DCs that were not stimulated or antigen-loaded (DC empty). There were significantly more neutrophils in the draining lymph nodes of mice one day following injection with DCs stimulated with LPS, regardless of antigen pulsing, than in the draining lymph nodes of mice that received DCs that were not stimulated with LPS (Figure 5). 

The communications Involved in the recruitment of neutrophils to the DC vaccine-draining lymph node were investigated further to evaluate if this process required cell contact or was cell contact-independent. The supernatant of mature and immature *ex vivo* generated DCs as well as the culture media as a control, were injected into the footpads of mice. The draining lymph nodes were then examined for neutrophil recruitment four hours post-injection. There was a significant increase in the number and proportion of neutrophils in the draining lymph nodes of the mice that received the mature DC-derived supernatant compared to those that received the immature DC-derived supernatant or media alone (Figure 6). 

Next, using a neutrophil-depleting antibody the effect neutrophils could have on the migration of the DC vaccine to the local draining lymph node was investigated to begin to evaluate the functional role neutrophils play in the immunobiology behind this DC vaccination. The DC vaccine was labeled with the fluorescent dye, carboxyfluorescein succinimidyl ester (CFSE), to allow for tracking of the vaccine using flow cytometry. A higher recovery of the DC vaccine in the footpad and draining lymph node of mice with neutrophils was observed compared to the mice depleted of neutrophils (Figure 7). 

Since we observed that neutrophils influenced the migration of the DC vaccine to the local draining lymph node, we next sought to evaluate if neutrophils could be influencing the immune responses occurring in the lymph node. We started by evaluating the effect of depleting neutrophils on NK cell responses following DC vaccination. NK cells have cytotoxic capacity and contribute to direct tumor cell killing, making them important effector cells in cancer biology. Furthermore, the established relationship between DCs and NK cells has important roles and implications in the immunity afforded by DC vaccines [6,22]. 

To begin investigating the influence of neutrophils on DC vaccine-mediated priming of NK cell responses, changes in the numbers of NK cells were investigated in the lymph node draining the DC vaccination site one day following immunization, as well as in naïve mice, and neutrophil-depleted vaccinated mice. Depletion of neutrophils reduced the total number and percentage of NK cells in the local vaccine-draining lymph node following DC immunization (Figure 8). 

Since depletion of neutrophils reduced the number and frequency of NK cells in the draining lymph node one day post-DC immunization, the functionality of the NK cells was investigated. From the kinetics experiment examining changes in cellularity in the local draining lymph node following DC immunization (Figure 1), two days post-DC immunization appeared to be an appropriate time to assess the impact of neutrophils on NK cell responses. This is because the number of NK cells in the lymph node peaked at this time. Therefore, the effect of neutrophils on the functionality of NK cells was assessed in draining lymph nodes two days post-DC immunization. Functionality was measured by the production of the effector cytokine interferon (IFN)γ and surface expression of CD107a, which is an indicator of degranulation [23]. Similar to one day post-DC immunization (Figure 8), neutrophil-depleted DC-immunized mice had decreased numbers of NK cells in the draining lymph nodes two days post-DC immunization compared to non-depleted DC-immunized mice (Figure 9). Upon stimulation with interleukin (IL)-2, there were significantly higher numbers of NK cells from both DC-immunized groups producing IFNγ and CD107a than their PBS counterparts. Furthermore, the DC-immunized mice had an increase in the total number of NK cells producing IFNγ and CD107a compared to the neutrophil-depleted DC-immunized mice. 

Since depletion of neutrophils reduced the functionality of DC vaccine-induced NK cells in the local draining lymph node, the effect of neutrophils on systemic NK cell responses was evaluated. To examine systemic changes, the splenic NK cell responses were assessed one-week post-DC immunization to allow sufficient time for systemic changes to occur. The DC-immunized mice had higher total numbers of NK cells in the spleen compared to control mice that received PBS, as well as the neutrophil-depleted DC-immunized mice. However, there were no significant differences between degranulation as measured by surface expression of CD107a or IFNγ production in DC-immunized mice with or without neutrophils depleted (Figure 10).

To further investigate the potential roles that neutrophils play in the immune responses elicited by the DC vaccine, CD8+ T cell responses were evaluated. The education of antigen-specific CD8+ T cells is a critical feature of DCs that is taken advantage of in the DC vaccine platform. Therefore, the spleens of DC-immunized mice were examined for CD8+ T cell responses to antigen stimulation with SIIN and ISQ. DC-immunization significantly increased the total number and proportion of splenic CD8+ T cells expressing CD107a, and producing IFNγ alone, or both IFNγ and tumor necrosis factor (TNF)α. However, there were no statistically significant differences in antigen-specific CD8+ T cell responses between the DC immunized group and the neutrophil-depleted DC-immunized group, in terms of numbers and proportions of degranulating and cytokine-producing T cells. In the presence of neutrophils, multi-cytokine-producing antigen-specific CD8+ T cells produced only slightly more IFNγ on a per cell basis, as indicated by higher geometric means in flow cytometry analysis (Figure 11). 

## 4. Discussion

DC vaccination influenced local draining lymph node cellularity (Figure 1). This opened an avenue to investigate how changes in cellularity can be manipulated to optimize DC vaccines to improve their efficacy. A population of neutrophils was observed in the draining lymph nodes shortly after the DC immunization but was absent in non-vaccine draining lymph nodes, suggesting the recruitment and/or accumulation of neutrophils is a localized event restricted to the DC vaccine-draining lymph node (Figure 4). Moreover, accumulation of neutrophils in the vaccine-draining lymph nodes required inoculation with LPS-matured DCs (Figure 5). This suggests that crosstalk between *ex vivo*-generated DCs and host neutrophils, which leads to accumulation of neutrophils in the lymph node draining the injection site requires the DCs to be in a mature state. Furthermore, there were significantly more neutrophils recruited into the draining lymph node following injection of supernatant from mature DCs than administration of supernatant from immature DCs or control media (Figure 6). These results indicate that *ex vivo*-cultured mature DCs can induce local recruitment of host neutrophils in a cell contact-independent mechanism. However, it should be noted that this does not rule out the possibility that a cell-to-cell contact-dependent recruitment mechanism may be occurring as well.

Functional roles for neutrophils in the DC vaccine immunobiology were investigated using a Ly6G-specific antibody to deplete neutrophils. There was a reduced recovery of the DC vaccine observed at both the injection site and the draining lymph node in the neutrophil-depleted mice compared to the non-depleted mice (Figure 7). This could imply roles for neutrophils in either supporting the migration of the DC vaccine to the lymph node and/or enhancing the survival of the DC vaccine in the host.

The influence of neutrophils on NK cells was also evaluated since they are implicated to be key players in DC vaccine immunobiology. There was an increase in the number and proportion of NK cells in the draining lymph node one day following DC immunization (Figure 8), that was diminished in Ly6G-depleted mice. This suggests that neutrophils can influence the well-established NK-DC crosstalk in DC vaccination. Moreover, there were fewer NK cells expressing CD107a and producing IFNγ upon *ex vivo* stimulation with IL-2 in the lymph nodes two days post-DC immunization in the neutrophil-depleted mice compared to the non-depleted DC-immunized mice (Figure 9). This suggests that neutrophils influence NK cell recruitment and/or accumulation in the lymph node and the magnitude of NK cytolytic activities (CD107a) and contribute to promoting type 1 immune responses (IFNγ). These observations indicate an important role of neutrophils in DC vaccination that quantitatively and functionally modulates NK cell responses.

Similar to the draining lymph nodes (Figure 8 and Figure 9), neutrophil depletion also appeared to negatively affect the total number of NK cells in the spleen following DC immunization. The spleens of DC-immunized mice had the highest number of NK cells compared to PBS-treated, neutrophil-depleted and PBS-treated, and neutrophil-depleted DC-immunized mice. However, there was no difference in the total number of NK cells expressing CD107a or producing IFNγ in the spleen of DC immunized mice compared to neutrophil-depleted DC immunized mice (Figure 10). Therefore, the influence of neutrophils on NK cell functionally following DC immunization observed locally two days post-DC immunization, was not observed systemically by seven days post-DC immunization. Overall, the data are suggestive of complicated crosstalk between the DC vaccine, neutrophils, and NK cells that could have implications for the immunobiology of DC-based cancer vaccines. 

Additionally, due to the importance of T cell education in DC vaccination platforms, the influence of neutrophils on the DC vaccine education of CD8+ T cell responses was evaluated. Ovalbumin was used as a well-characterized model antigen. Splenic antigen-specific CD8+ T cell responses of the DC-immunized group, and the neutrophil-depleted immunized group was significantly higher than both the PBS-treated and neutrophil-depleted PBS-treated control groups (Figure 11). This demonstrated that DC-based immunization could educate CD8+ T cell responses. However, there were little to no differences between the DC-immunized group and the DC-immunized mice that had neutrophils depleted. The only exception was that the most functional, multi-cytokine-producing T cells had slightly more IFNγ per cell. Therefore, neutrophils did not seem to have any substantial biological effect on the ability of DCs to induce CD8+ T cell responses, as per the indicators that were assessed in this study. 

NK cell and neutrophil crosstalk has been shown to activate DC maturation and promote adaptive T cell responses [24]. It was observed in vitro that activated neutrophils could stimulate NK cells to have enhanced crosstalk with DCs and activate DCs to promote more robust adaptive T cell responses than NK cells that were not conditioned with neutrophils [24]. Although no detectable differences in CD8+ T cell responses were observed here, there were observations of potential crosstalk between neutrophils and NK cells. These communications could have influences on T cell responses that were not captured by the experiments reported in this paper. Therefore, further investigations into the communications between neutrophils and NK cells and the influence on adaptive T cell responses in the context of DC vaccination is recommended. 

Neutrophils display either an N1 or N2 phenotype. N1 is associated with type 1 immunity, and N2 is associated with type 2 immunity [21]. N1 and N2 neutrophils represent immunostimulating and immunosuppressive subsets, respectively, in cancer. Neutrophils support metastasis [25,26], T cell exhaustion [27] and angiogenesis [28]. They have also been implicated in protecting tumor cells [26,29], helping them travel and seed in other locations. Furthermore, in the tumor microenvironment, there is evidence that they produce neutrophil extracellular traps which block and protect cancer cells from cytotoxic effector cells, NK and T cells [30]. These functions could be attributed to the pro-tumorigenic phenotype that neutrophils can obtain. However, neutrophils were also shown to have functional capacities that could support anti-tumor immunity [31,32]. They contribute to the formation of adaptive immunity, like acting as a source of antigens for DCs [33] or directly presenting antigens to T cells [11,34]. Neutrophils have also been observed to recruit and activate DCs [35,36], which could support anti-tumor immunity. These functions that promote immunogenic responses could be attributed to anti-tumor neutrophils (N1s). Therefore, cancer immunotherapies could be designed to target neutrophils and promote N1 phenotypes and suppress N2 phenotypes. N1 and N2 cells have demonstrated plasticity that can be stimulated with DC-derived cytokines such as tumor growth factor-β or type I IFNs [37,38,39]. Thus, DCs represent a way to influence neutrophil phenotypes. Therefore, through optimizing DC vaccine generation, there is potential to enhance anti-tumor responses elicited via manipulation of neutrophil phenotypes. However, to pursue this avenue of research, the communications between DC vaccines and neutrophils must be fully characterized first.

The advancements in transcriptomic analysis techniques would be beneficial and allow for examining complex phenotypic diversity through genomic expression. Exploring and analyzing the phenotypic and functional diversity of neutrophils and their responses in such circumstances, as well as the influence on the genomic expression in the other leukocytes, such as CD8+ T cells and NK cells, could dissect cellular responses downstream of DC vaccination.

The data presented here demonstrate a key role for neutrophils in the formation of immune responses elicited by DC vaccination, especially in the context of quantitatively and qualitatively enhancing NK cell responses. However, further investigation into the influence neutrophils have on tumor burden, overall survival, and anti-tumor responses would help elucidate the communications and crosstalk occurring and the influence on the vaccine immunobiology. A better understanding of the immunobiology underpinning this immunization technology will allow for the development of enhanced DC vaccine strategies.

## Figures and Tables

**Figure 1 cells-12-00121-f001:**
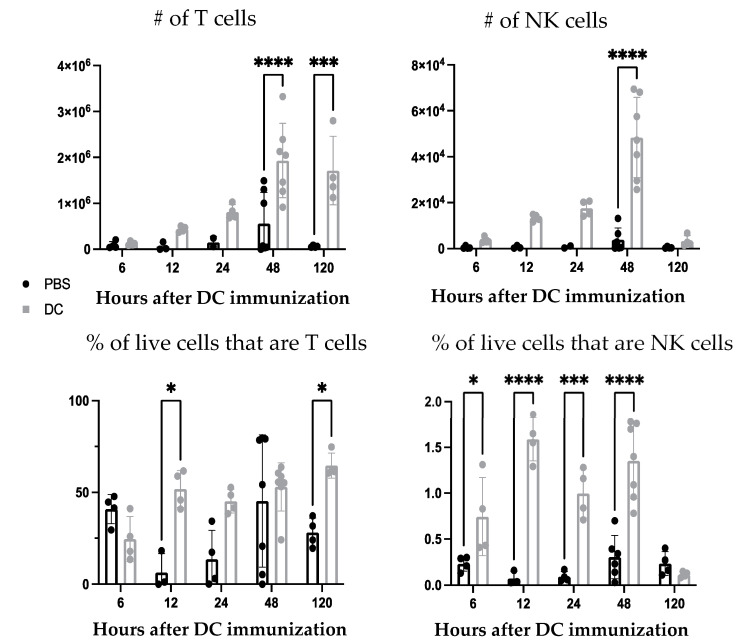
**The cellularity of the local draining lymph node changed following dendritic cell (DC)-based vaccination.***Ex vivo*-generated DC-based vaccines were administered into the hind footpads of female C57Bl/6 mice (n = 2–7 per treatment), and then the popliteal lymph nodes were examined six-, 12-, 24-, 48-, and 120 h post-immunization to evaluate changes in cellularity. The number and percentage of natural killer (NK) and T cells in the lymph nodes were quantified at each time point following DC immunization using flow cytometry and compared with lymph nodes from control mice that received injections of phosphate-buffered saline (PBS). Statistical analysis was performed using a two-way analysis of variance; *p*-value < 0.05 (*), *p*-value < 0.001 (***), *p*-value < 0.0001 (****). Graphs show the means and the standard deviation with each dot representing an individual mouse.

**Figure 2 cells-12-00121-f002:**
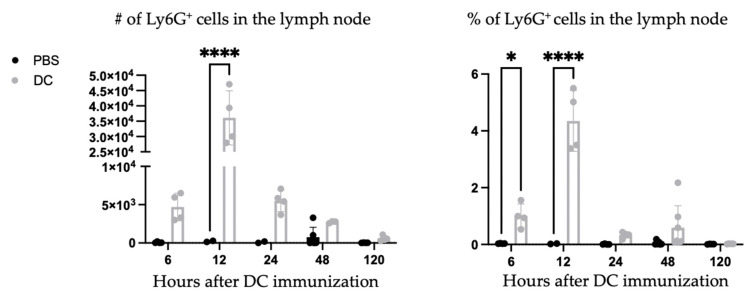
**Rapid recruitment of a Ly6G^+^ population to the local draining lymph node following dendritic cell (DC) vaccination.***Ex vivo*-generated DC vaccines were injected into the hind footpads of female C57BL/6 mice (n = 2–6 per treatment), and the local draining lymph node was examined for Ly6G^+^ cells. Ly6G+ cells were assessed by flow cytometry six, 12, 24, 48, and 120 h post-DC immunization. The number and percentage of Ly6G^+^ cells in the lymph node of DC-immunized mice were compared with phosphate-buffered saline (PBS)-injected control mice. Significance was determined statistically using a two-way analysis of variance.; *p*-value < 0.05 (*), *p*-value < 0.0001 (****). Graphs show the mean and the standard deviation with each dot representing an individual mouse.

**Figure 3 cells-12-00121-f003:**
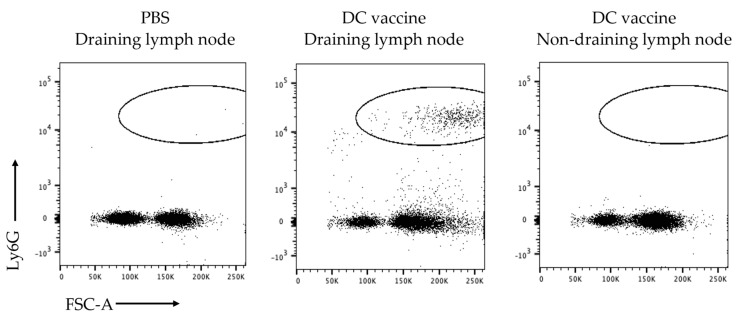
**Ly6G^+^ cell recruitment was limited to the vaccine-draining lymph node.** Dendritic cell (DC) vaccines were injected into one hind footpad of each female C57BL/6 mouse and phosphate-buffered saline (PBS) into the other hind footpad. Both popliteal lymph nodes were examined for Ly6G^+^ populations 12 h post-immunization, one representing the PBS-draining lymph node (**left panel**) and the other as a local vaccine-draining lymph node (**middle panel**). The inguinal lymph node on the vaccine-administered side of the mouse was also examined as a non-vaccine-draining lymph node (**right panel**). Representative flow cytometry dot plots of live cells in the lymph nodes are shown.

**Figure 4 cells-12-00121-f004:**
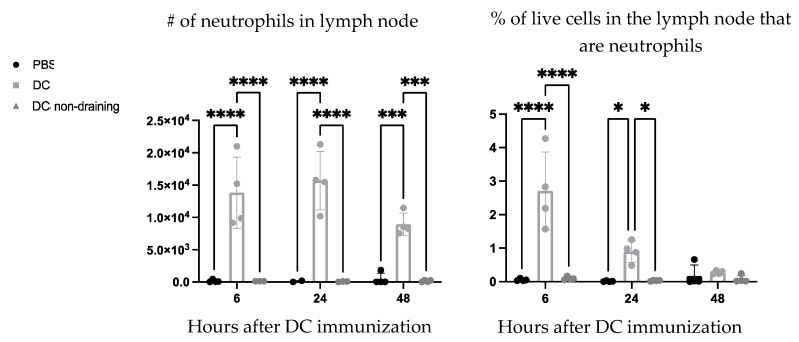
**Neutrophils were recruited to the dendritic cell (DC) vaccine-draining lymph node.** DC vaccines were generated *ex vivo* and administered into the hind footpads of female C57BL/6 mice (n = 2–4 per treatment). The vaccine-draining popliteal lymph nodes (labeled “DC”) were examined for neutrophils (Ly6G+ CD11b+) six-, 24-, and 48 h following DC vaccination and were compared with non-draining inguinal lymph nodes from the vaccinated side of mice (“DC non-draining”), and phosphate-buffered saline- draining lymph nodes (“PBS”). Statistical analysis was performed using a two-way analysis of variance; *p*-value < 0.05 (*), *p*-value < 0.001 (***), *p*-value < 0.0001 (****). Graphs show the means and the standard deviation with each dot representing an individual mouse.

**Figure 5 cells-12-00121-f005:**
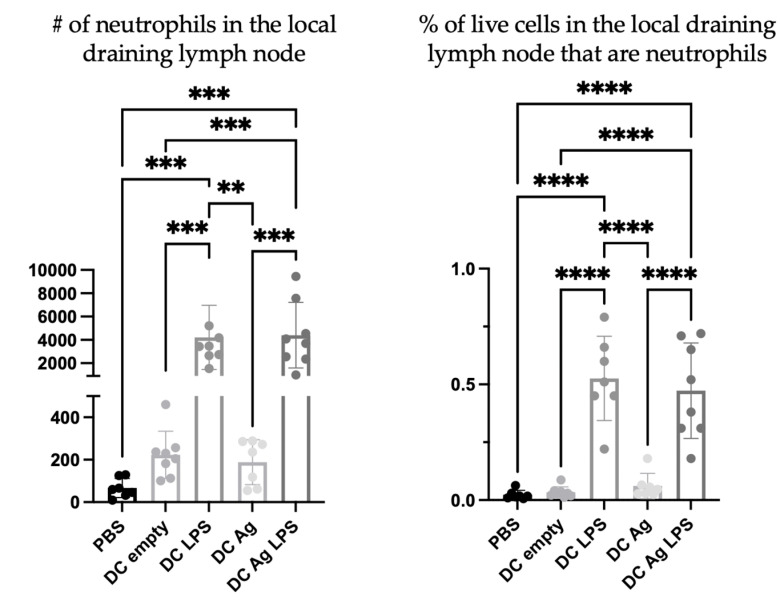
**Neutrophil accumulation in the draining lymph nodes required maturation of dendritic cells (DCs).** DC vaccines were prepared using DCs cultured *ex vivo*. The DCs were matured using lipopolysaccharide (LPS) as a source of pathogen-associated molecular pattern signaling that induces activation and maturation of DCs, and then they were loaded with antigens. DC cultures were prepared for immunization using different iterations of the manufacturing protocol. These included DCs stimulated with LPS only (DC LPS), DCs pulsed with antigens only (DC Ag), DCs pulsed with antigens and stimulated with LPS (DC Ag LPS), and DCs without LPS treatment or antigen-pulsing, which were termed “DC empty”. The number and percentage of neutrophils (Ly6G^+^ CD11b^+^) in the draining lymph nodes of the female C57BL/6 mice were examined one day following inoculations using flow cytometry. The data were analyzed for significance using a one-way analysis of variance; *p*-value < 0.005 (**), *p*-value < 0.001 (***), *p*-value < 0.0001 (****). Graphs show the means and the standard deviation with each dot representing an individual mouse.

**Figure 6 cells-12-00121-f006:**
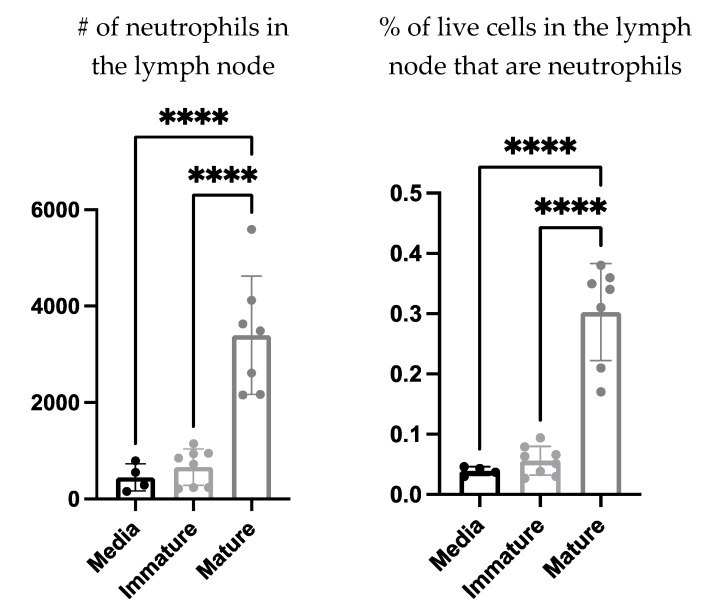
**Soluble factor(s) produced by mature dendritic cells (DCs) were capable of recruiting neutrophils to the local draining lymph node.** DCs were cultured *ex vivo* and half of the culture was matured using lipopolysaccharide, and the other half remained in an immature state. The DCs were plated in media and left overnight in an incubator at 37 °C. The supernatants of the immature and mature DCs were collected and injected into the hind footpads of female C57BL/6 mice (n = 4–7 per treatment), and media only was injected into negative control mice. Four hours post-injection, the draining lymph nodes were examined for neutrophils (Ly6G^+^ CD11b^+^). Significant differences were examined using a one-way analysis of variance; *p*-value > 0.0001 (****). Graphs show the means and the standard deviation with each dot representing an individual mouse.

**Figure 7 cells-12-00121-f007:**
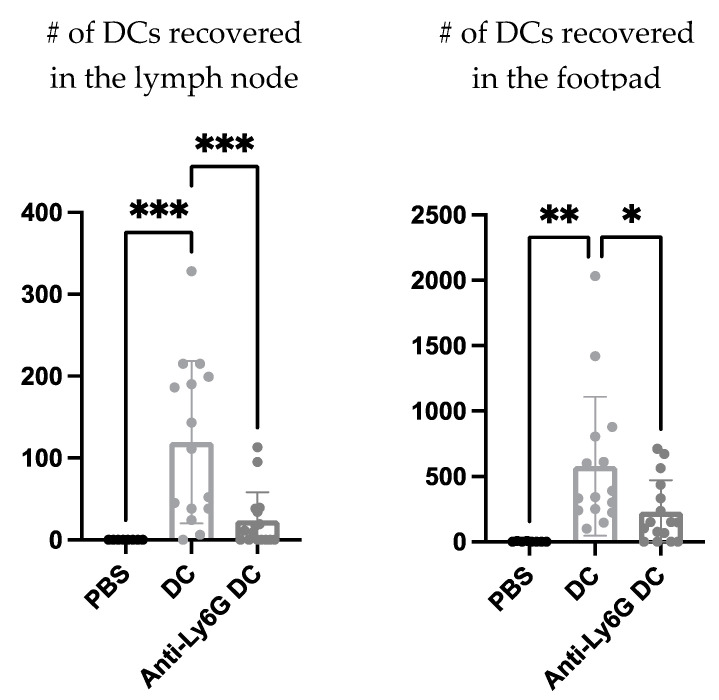
**Neutrophils influenced dendritic cell (DC) vaccine recovery in the footpad and draining lymph node one day post-immunization.** DC vaccines were prepared *ex vivo* and labelled with the fluorescent dye, carboxyfluorescein succinimidyl ester (CFSE). Female C57BL/6 mice (n = 8–16 per treatment) were treated with an anti-Ly6G antibody one day prior to and on the day of DC immunization to deplete neutrophils from the mice. The footpads and draining lymph nodes were examined for DC vaccine recovery at the injection site and at the draining lymph node one day following inoculation to evaluate the influence of neutrophils on the migratory capacities of the vaccine. The numbers of DCs in these tissues from mice that received anti-Ly6G and the DC vaccine were compared to mice that received the DC vaccine only, and negative control mice that received phosphate-buffered saline (PBS). Significant differences were determined using a one-way analysis of variance; *p*-value < 0.05 (*), *p*-value < 0.005 (**), *p*-value < 0.001 (***). Graphs show the means and the standard deviation with each dot representing an individual mouse.

**Figure 8 cells-12-00121-f008:**
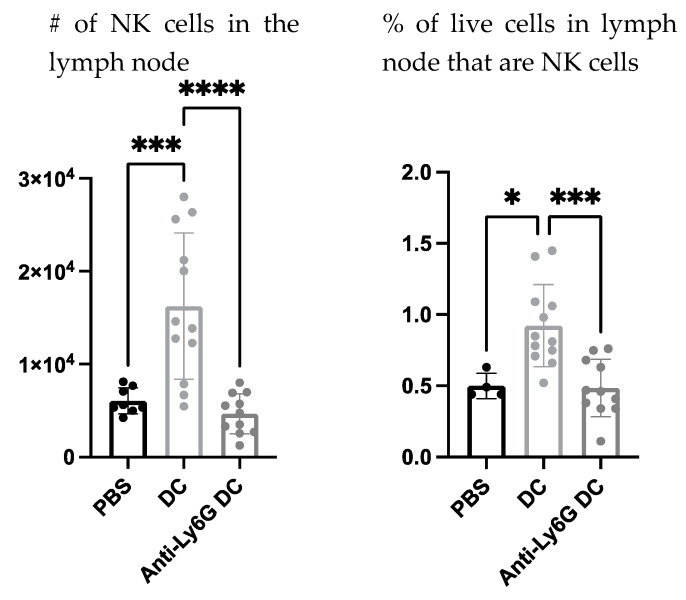
**Depletion of neutrophils influenced the accumulation of natural killer (NK) cells in the local draining lymph node one day following dendritic cell (DC) immunization.** Anti-Ly6G was administered to female C57BL/6 mice (n = 4–12 per treatment) on the day before and the day of DC immunizations to deplete neutrophils. DC vaccines were prepared *ex vivo* and administered into the footpads of mice. One day post-DC immunization, the local draining lymph nodes were examined by flow cytometry for NK cell populations. Statistical analysis was performed using a one-way analysis of variance; *p*-value < 0.05 (*), *p*-value < 0.001 (***), *p*-value < 0.0001 (****). Graphs show the means and the standard deviation with each dot representing an individual mouse.

**Figure 9 cells-12-00121-f009:**
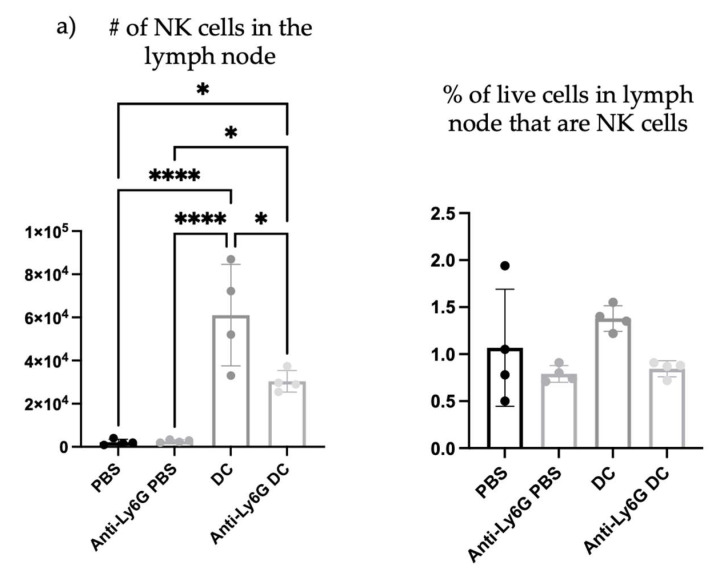
**Neutrophils influenced the functionality of natural killer (NK) cells in local draining lymph nodes two days following dendritic cell (DC) immunization.** Half of the female C57BL/6 mice received anti-Ly6G to deplete neutrophils the day before and the day of immunizations, and then DC vaccines or phosphate-buffered saline (PBS) were administered (n = 4 per treatment) into hind food pads. (**a**) Two days later, the number and proportion of NK cells in the draining popliteal lymph nodes were compared between neutrophil-depleted and non-depleted mice. (**b**) NK cells were stimulated using IL-2, and the production of IFNγ (left panels) and CD107a (right panels) were measured using flow cytometry. A two-way analysis of variance was used to assess statistical significance; *p*-value < 0.05 (*), *p*-value < 0.005 (**), *p*-value < 0.001 (***), *p*-value < 0.0001 (****). Graphs show the means and the standard deviation with each dot representing an individual mouse.

**Figure 10 cells-12-00121-f010:**
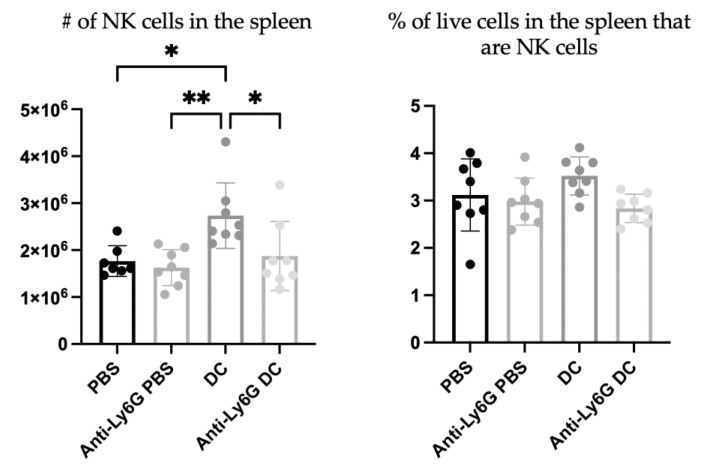
**Influence of neutrophils on natural killer (NK) cell responses in the spleen one-week post-dendritic cell (DC) vaccination.** Anti-Ly6G was given to half of the female C57BL/6 mice to deplete neutrophils (n = 8 per treatment). DC vaccines generated *ex vivo* were administered. Controls received phosphate-buffered saline (PBS). One week later, NK cell responses in the spleens were assessed. Splenocytes were stimulated with interleukin (IL)-2, and NK cell degranulation based on expression of CD107a and production of IFNγ were measured using flow cytometry. A two-way analysis of variance was used to evaluate statistical significance; *p*-value < 0.05 (*), *p*-value < 0.005 (**), *p*-value < 0.0001 (****). Graphs show the means and the standard deviation with each dot representing an individual mouse.

**Figure 11 cells-12-00121-f011:**
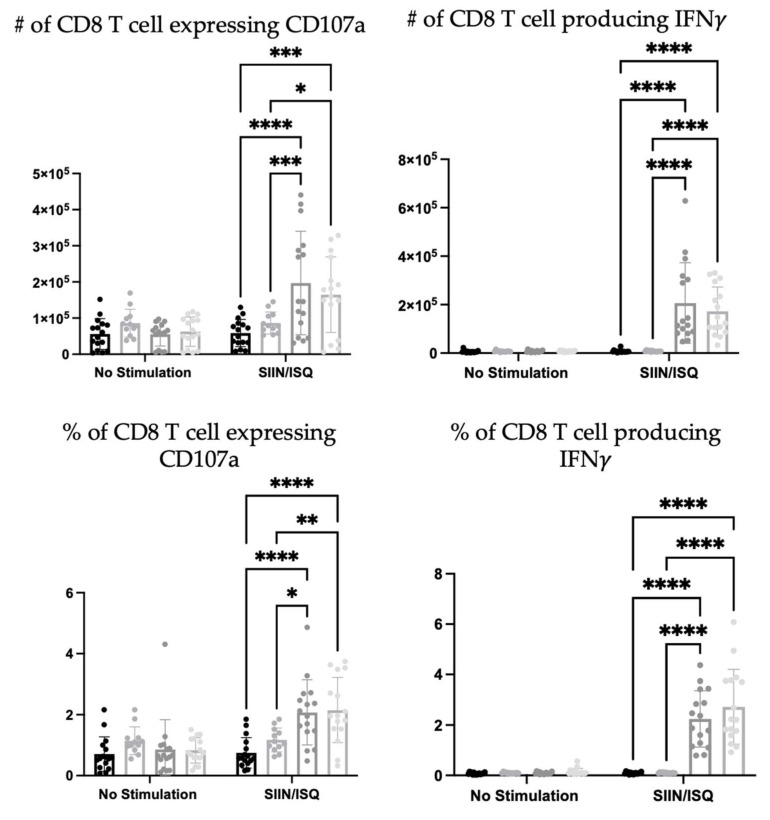
**Neutrophils did not influence dendritic (DC) vaccine-mediated education of antigen-specific CD8+ T cell degranulation or production of interferon (IFN**γ**) and tumor necrosis factor (TNF)**α**.** Female C57BL/6 mice (n = 12–16 per treatment) were given anti-Ly6G followed by DC vaccines. One week later, the spleens were examined for CD8+ T cell responses to stimulation with the vaccine-loaded epitopes, ovalbumin (OVA)_257-264_ (SIIN), which is the immunodominant CD8+ T cell epitope, and OVA_323-339_ (ISQ), which is the dominant CD4+ helper T cell epitope. The CD8+ T cell responses were measured by expression of CD107a as an indicator of degranulation, and production of IFNγ and TNFα using flow cytometry. The CD8+ T cell responses in the spleens of mice that received anti-Ly6G and the DC vaccine, the DC vaccine only, PBS only, and anti-Ly6G and PBS were compared using a two-way analysis of variance; *p*-value < 0.05 (*), *p*-value < 0.005 (**), *p*-value < 0.001 (***), *p*-value < 0.0001 (****). Graphs show the means and the standard deviation with each dot representing an individual mouse.

## Data Availability

Data are available from the corresponding author upon reasonable request.

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
