# Peer review of "Dendritic Cell-Based Vaccines Recruit Neutrophils to the Local Draining Lymph Nodes to Prime Natural Killer Cell Responses"

_cells, 2022, doi:10.3390/cells12010121_

Round 1

Reviewer 1 Report

This research is based on the good effect of dendritic cell based cancer vaccine, and explores the role of neutrophils in its function. This research has a clear idea, a full workload and certain clinical significance from the superficial findings to functional research and from local to systemic research. The findings of this study can provide innovative strategies for DC vaccine anti-tumor immunity to improve its clinical efficacy.

Reviewer 2 Report

The manuscript of Chan L et al. elegantly shows, by a flow cytometry analysis, that a population of neutrophils infiltrate the LNs soon after the DC vaccination and influence the resulting immune response, mainly affecting NK cell functionality. Neutrophils were recently shown to be functionally versatile and perform unexpected functions, including activation of other innate responses and involvement in the education of adaptive immunity. Therefore, this manuscript explores a critical theme still unexplored in immune responses, and the approach used by the authors is well-designed and clearly explained.

However, some aspects of the flow-cytometry experiments are not apparent:

1. The gating strategy used to identify the cell population should be shown. I found it strange that the authors do not use CD45 to determine the total amount of leukocytes. It is frequently used to express the amount of a specific cell population on the total number of leukocytes.

2. Did the authors obtain the same amount of cells in LNs from the different experimental conditions or in draining versus not draining LNs? I understand that they always label 1 million cells in each condition, but the global number in LNs also indicates the immune response in that site. In addition, as the cells are obtained manually from different tissues, there is a high risk of losing cells in some cases and getting more in others. The authors should present raw numbers of the LNs cellularity in the experiments; otherwise, before giving the raw number of the cells or percentages, they have to normalize the total number of the cells obtained.

3. Did they see variabilities among the availability of cells in the different experimental conditions?

4. How did they calculate the % of each population in Figures 1 and 2?

In Figures 6, 7, and 8, there is a high variation among mice in the DC condition; do the authors have an explanation?

Regarding the data, why do the authors show in Figure 1 the # and % of NK cells, NKT cells, and T cells instead of showing CD4 and CD8 separately? I am asking that, considering that they focus only on CD8+ cell responses later in the manuscript. Please, show all the lymphoid subtypes individually or only the CD8 cells for the sake of clarity.

Figure 1 also shows the increased # of NKT cells at 48h after DC immunization, in agreement with T and NK cell responses; why have the authors not considered exploring this cell population after DC and Ly6G-depleted DC experiments? Consider removing this info if you do not want to examine the effect of neutrophil depletion on NKT cells. As an alternative, please comment on that.

For clarity, the authors could also include a schematic diagram of the experimental procedure used in neutrophil depletion experiments.

Did the authors consider exploring the phenotype of neutrophils? This can be quickly done by flow cytometry and will be helpful to guess what their role is in immune responses elicited by DC vaccination.

Did the authors verify that the anti-Ly6G antibody was depleting the neutrophils from the peripheral circulation?

In figure 5, the authors show that neutrophil accumulation in the draining lymph nodes requires the maturation of dendritic cells. Could it be that some LPS used to mature DCs was injected together with the cells? This would also explain data regarding the supernatant of mature and immature ex-vivo-generated DCs that were injected into the footpads of mice. As a control, did the authors try injecting LPS only into mice’s footpads to see if neutrophil infiltration would occur in draining LNs?

In Figure 10, the authors attempt to show the systemic response of neutrophil depletion in NK cells isolated from the spleen. However, no differences are shown in post-dendritic cell vaccination in neutrophil-depleting conditions after one week. Did they check other time points?

What implication would it have in the case of a vaccination protocol for cancer treatment? Do you then expect neutrophils somehow limit the systemic response, or, on the contrary, would it be helpful to manipulate neutrophils to obtain a specific and circumscribed response to the organ affected by the pathology? Please, comment on these findings to enrich the discussion.

The authors perform experiments using OVA immunodominant peptides for CD4 and CD8 but only consider the CD8 response. Is there any reason? What was the purpose of using the two peptides if the authors focused on CD8+T cells only? Since neutrophils migrating to LNs were recently shown to modulate CD4+ T cell responses, and CD4 is among the cell population that can be easily identified in their gating strategy, the authors should also show and comment on their findings about data present in the literature.

Reviewer 3 Report

Authors have submitted the "dendritic cell based immunotherapy that recruit the nutrophil to prime NK cells" research study which is good in current form.

The following queries need to be answered:

1) Authors are requested to add graphical presentation of whole experiment study with parameters in one frame so that it will be easy for readers.

2) Did authors observe any response of macrophage?

3) Trasncriptomic analysis in future would be the good add in parameters to know further.

Result is presented well and authors have critically discussed limitation and flaws in the discussion part.

 It would be interesting to see in future if any research group comes up with macrophage and dendritic cell cocktail based vaccine immunotherapy with prime-boost strategy or any appropriate strategy so that how the cells  respond can be elucidated and its role in cancer immunotherapy.
